# Effect of Wet–Dry Cycling on Properties of Natural-Cellulose-Fiber-Reinforced Geopolymers: A Short Review

**DOI:** 10.3390/molecules28207189

**Published:** 2023-10-20

**Authors:** Chun Lv, Pengyi He, Guowei Pang, Jie Liu

**Affiliations:** 1College of Architecture and Civil Engineering, Qiqihar University, Qiqihar 161006, China; 2College of Light-Industry and Textile Engineering, Qiqihar University, Qiqihar 161006, China; 3Engineering Research Center for Hemp and Product in Cold Region of Ministry of Education, Qiqihar 161006, China

**Keywords:** NCF, NCFRG, bonding property, long-term property, mechanical property

## Abstract

To study the long-term properties of cement-based and geopolymer materials exposed to outdoor environments, wet–dry cycles are usually used to accelerate their aging. The wet–dry cycling can simulate the effects of environmental factors on the long-term properties of the composites under natural conditions. Nowadays, the long-term properties of geopolymer materials are studied increasingly deeply. Unlike cement-based materials, geopolymers have better long-term properties due to their high early strength, fast hardening rate, and wide range of raw material sources. At the same time, natural cellulose fibers (NCFs) have the characteristics of abundant raw materials, low price, low carbon, and environmental protection. The use of NCFs as reinforcements of geopolymer matrix materials meets the requirements of sustainable development. In this paper, the types and properties of NCFs commonly used for geopolymer reinforcement and the polymerization mechanism of geopolymer matrix materials are summarized. By analyzing the properties of natural-cellulose-fiber-reinforced geopolymers (NCFRGs) under non-wet–dry cycles and NCFRGs under wet–dry cycles, the factors affecting the long-term properties of NCFRGs under wet–dry cycles are identified. Meanwhile, the degradation mechanism and mechanical properties of NCFRG composites after wet–dry cycles are analyzed. In addition, the relationship between the properties of composites and the change of microstructure of fiber degradation is further analyzed according to the results of microscopic analysis. Finally, the effects of wet–dry cycles on the properties of fibers and geopolymers are obtained.

## 1. Introduction

In today’s world construction field, Portland cement is still the most widely used and the largest consumption of cementitious material. However, the production of this material consumes a lot of energy and resources and emits a lot of carbon dioxide, which will contribute to the greenhouse effect. It hurts reducing carbon emissions and achieving carbon neutrality. Therefore, reducing the production and use of cement and finding cement substitutes has become a research hotspot in the field of construction [1]. Among them, geopolymers are considered as a green building materials to replace traditional Portland cement due to its low energy consumption, low carbon emissions, and superior mechanical properties compared with traditional cement [2,3].

Geopolymers are formed by polymerization of an active silica-aluminum material with an alkaline activator solution. The synthesis of geopolymers requires an active solid silicoaluminate precursor and alkaline activator solution. Alkaline activator solution has the function of binder, activator, and dispersant [4]. Polymeric aluminum silicate material was formed by exciting geological minerals with an alkali metal silicate solution under strong alkaline conditions. The inorganic silica-alumina cementing material has a three-dimensional network structure. The commonly used active silicate raw materials include fly ash [5,6], refined blast furnace slag [7] and other wastes, such as red mud, rice husk ash, and some mine tailings [8,9,10]. All kinds of geopolymers with excellent properties have been prepared.

Plant fibers, also known as vegetable fibers, are natural celullose fibers (NCFs). Compared with other traditional fibers, NCFs have the characteristics of low price, good mechanical properties, good adhesion to matrix, and biodegradability [11,12]. Due to people’s increasing concern about environmental issues, engineering researchers are trying to find sustainable and can replace traditional synthetic fibers as structural reinforcement fibers. Based on this, natural cellulosic fibers, such as sisal, jute, cotton, flax, hemp, and various other straws, have emerged as potential alternatives [13]. This is because these fibers are good and environmentally friendly, and can be extracted at a low cost from plant stems, leaves, and fruits. Natural-cellulose-fiber-reinforced geopolymer (NCFRG) composites have strong environmental advantages, reducing dependence on non-renewable resources, reducing pollutant emissions, reducing greenhouse gas emissions, and improving energy recovery [14].

Building materials often suffer from the natural environment because they are exposed to various outdoor climatic conditions. The application of traditional cement-based composites is limited due to their sensitivity to weather resistance. The properties of geopolymer materials can overcome many performance defects of cement-based materials [15]. However, similar to cement-based materials, geopolymers are brittle materials. Although its compressive strength is good, its tensile strength is weak and it is prone to micro-cracks [16,17]. Of course, the ductility of the composite can be improved by incorporating various fibers into the matrix. The addition of fiber into the geopolymer matrix can enhance the toughness of the geopolymer and further limit the propagation of microcracks [18]. In the past 20 years, many scholars have studied the long-term properties of geopolymers. Such studies mainly focus on the properties of geopolymers such as sulfate corrosion resistance, freeze–thaw resistance, weathering resistance, chemical erosion resistance, and chloride ion resistance [19,20]. Research on the long-term properties of cement-based and geopolymer materials exposed to outdoor environments usually adopts the method of wet–dry cycling to accelerate their aging [21,22]. The wet–dry cycle method can simulate the natural environment to accelerate the aging of composite materials and carry out research on the long-term properties of composite materials under natural conditions [23]. By adjusting the mixture ratio of precursor materials, the types and dosage of different activators, different curing conditions, and adding various kinds of fibers, nanomaterials, and other admixtures, the long-term performance of the materials under the condition of wet–dry cycle can be improved. Especially, the addition of fiber improves the tensile strength and fracture behavior of the composite. Studies have shown that fiber-reinforced geopolymers have better long-term properties compared with fiber-reinforced cement-based materials [24,25].

In this paper, the factors affecting the mechanical properties of NCFRG composites were summarized and analyzed based on the relevant literature on the influence of wet–dry cycling tests on the properties of cellulose-fiber-reinforced geopolymer composites in recent years and the properties of NCFs. The types and properties of NCFs commonly used for geopolymer reinforcement were summarized. The composition and polymerization mechanism of geopolymer matrix materials were briefly analyzed. The factors influencing the long-term properties of NCFRG composites under the condition of wet–dry cycles were summarized. The degradation mechanism and mechanical properties of NCFRG composites after repeated wetting and drying were studied. In addition, according to the results of microscopic analysis, the relationship between the structure and properties of the materials and the microstructure changes of the fiber degradation were analyzed. Finally, the development direction of NCFRG materials is proposed.

## 2. Classification and Properties of NCFs

NCFs are derived from plant fibers, also known as lignocellulose fibers or vegetable fibers. Its main components include cellulose, hemicellulose, and lignin [26]. There are many kinds of NCFs, which is one of the most abundant natural resources. The mechanical and physical properties of different types of NCFs are different, and the properties of different types of NCFs are shown in Table 1.

As can be seen from Table 1, the density of NCFs is relatively small, the unit mass is also relatively light, and it has good tensile strength. Compared with other types of fibers, NFC has a higher elongation at break. For example, the elongation at break of steel fiber is 4.2%, and the elongation at break of carbon fiber is even lower, which is only 1.5% [37]. When mixed with cement materials and geopolymer materials, it is also more evenly distributed in the matrix [37].

NCFs are widely found in agricultural residues such as straw, wheat, corn, sorghum straw, also including bagasse, wood chips, shavings, etc. [38]. According to different types and parts of plants, NCFs commonly used for geopolymer reinforcement can be divided into bast fibers, leaf fibers, fruit fibers, seed fibers, and stem fibers [39]; their concise classification is shown in Figure 1.

According to Lv et al. [11], NCFs composed of cellulose, hemicellulose and lignin have unique microstructure. The chemical composition of NCFs from different sources is different, and the effect of NCFS on the interface adhesion between NCFs and the matrix is also significantly different, as shown in Table 2.

## 3. Properties of NCFRG Composite

The good properties of fiber provide conditions for improving the brittleness of the gelled material matrix. In particular, NCFs have abundant resources and low prices. NCFs are used together with geopolymers that are used to replace cement-based materials, in line with the concept of low carbon emissions, environmental protection, and sustainable development [43].

### 3.1. Types of Geopolymers

Figure 2 shows the main material composition of different geopolymer precursors and activators. According to the composition of precursor raw materials, geopolymers are mainly divided into two categories, which are alkali-silicate vitreous active materials and alkali-silicate mineral active materials. The active material of the alkali-silicate vitreous body is an amorphous silicate vitreous body, including slag, fly ash, red mud, and coal gangue [44,45,46]. The main raw materials of alkali-silicate mineral active materials are metakaolin, clay, feldspar, and other crystalline minerals [47,48].

### 3.2. Polymerization Mechanism of Geopolymers

As shown in Figure 2, the active materials of geopolymers mainly include silica-aluminous precursor materials and alkaline activator solutions. Strong alkali solutions act as activators in active silica-alumina materials. The polymerization process of geopolymers is shown in Figure 3. When the precursor active material is excited by a strong alkali, the silico-oxygen bond or alumino-oxygen bond breaks, and then forms a monomer oligomer tetrahedron. Then, the strong base continues to act on the oligosiloxy or aluminoxy tetrahedron, prompting it to dehydrate and polymerize to form a three-dimensional spatial network structure [49]. The geopolymerization process of geopolymers usually includes dissolution, diffusion, polymerization, and curing. The polymerization process can be described in detail as the precursor active material is gradually dissolved in the alkali activator to produce a large number of silicon and aluminum monomers. These silicon and aluminum monomers gradually diffuse in alkaline solutions from the surface to the inside. The oligomeric gel phase of silicon and aluminum is formed by condensation polymerization at the same time as the diffusion of silicon and aluminum monomers. After solidification and hardening, the polymerization process of geopolymers is finally completed, forming geopolymer concrete and releasing a certain amount of heat [50,51].

### 3.3. Bonding Properties of NCF and Geopolymers

As mentioned above, NCFRG composite material consists of a geopolymer matrix and NCF reinforcement. The bonding between matrix and fiber has a great influence on the properties of the composite. The bonding properties of matrix and fiber depend on their interfacial bonding states and bonding characteristics. By improving the interface bonding state and optimizing the bonding characteristics between the geopolymer matrix and the NCF, composite material can achieve improved bonding performance [52,53]. It can be said that improving the interface adhesion between the geopolymer matrix and NCF reinforcement is one of the key factors to improve the mechanical properties of composite materials. Relevant researchers believe that the interface between the geopolymer matrix and NCF can achieve good interface bonding, mainly through mechanical interlocking, chemical bonding, mutual diffusion, and electrostatic adhesion between the interfaces [54].

### 3.4. Water Absorption Properties of NCFRG

Building materials are often affected by natural environmental factors in the service process; thus, the composite material repeatedly undergoes wet–dry stages. In general, when the composite material absorbs water, the mechanical properties of the composite material decline. Water molecules can break the bond between fiber and matrix. The volume of NCF expands after absorbing water, which destroys the interface between fiber and matrix and affects the effective transfer of external force [55]. On the other hand, hydrogen bonds are formed between fibers and water molecules, and this hydrogen bond also breaks the bond between fibers and the substrate [56]. NCF-reinforced polymers have high water absorption and are more sensitive to water in the environment. The water absorption of NCF-reinforced polymer composites is related to the properties of the materials. Due to the large number of micropores inside the fiber-reinforced geopolymer composite, water molecules can diffuse into the interior of the material through these micropores [57]. The water absorption pathway of geopolymer matrix composites includes three parts. The first is diffusion within the matrix of the composite material, that is, water molecules can be diffused along the small gap between the molecular chains of the composite material. Secondly, the capillary effect of the micro gap between the interface of the composite material. Finally, water molecules are diffused in interfacial cracks caused by fiber expansion [58].

Through the above analysis, we found that the water absorption process of the composite material is mainly achieved through the diffusion of water molecules. The reason why water molecules can enter the interior of the composite material is that, on the one hand, the existence of micropores in the interior of the composite material causes the diffusion of water molecules from the area of high concentration to the area of low concentration through the internal micropores of the composite material. On the other hand, it is also related to the molecular polarity, the type, and the number of functional groups of the composite components.

The water absorption of composites is closely related to the properties of NCFs. NCF molecules contain a large number of hydroxyl groups, which can bind to water molecules. Mourak et al. [59] found that the presence of cellulose fiber in jujube trees reduced the porosity of the composite material, and the water absorption increased from 11.3 to 19%. The water absorption of different types of cellulose fibers is very different. More et al. [60] found through experiments that compared with other types of fiber, the maximum water absorption of coconut shell fiber could reach 180%. Under the same conditions, the maximum water absorption of jute fiber is only 40%. In addition, the water absorption of the same fiber is also different under different environmental conditions. Wei et al. [61] found that the water absorption of sisal fiber soaked in lime water was 5% higher than that of sisal fiber soaked in water. The researchers also found that water molecules penetrate NCFs to form hydrogen bonds, thus exposing the NCFs to water to create interfacial composites with the matrix material [62,63].

Under the influence of the wet–dry cycle, the NCF repeatedly absorbs water and expands and shrinks with water loss, which causes the stress generated inside the fiber and then forms micro-cracks [64]. Methacanon et al. [65] studied the hygroscopic properties of different NCFs under different relative humidity environments. The study found that water hyacinth fibers absorbed almost eight times more water at 97% relative humidity than at 75% relative humidity. However, when the relative humidity of the air increased to 97%, the moisture absorption rate of sisal, reed, and rose leaf fibers increased by only 4%. It can be seen that the hygroscopic properties of different types of NCFs are very different from the relative humidity of the environment.

## 4. Mechanical Properties of NCFRG under Non-Wet–Dry Cycle Conditions

The geopolymer matrix and cellulose fiber reinforcement determine the properties of geopolymer composites under wet–dry cycle conditions. The research on the ability of composite materials to resist the long-term damage of the natural environment includes the properties of acid erosion resistance, chloride ion penetration resistance, and high-temperature resistance of composite materials in addition to the influence of the wet–dry cycle [66]. Relevant studies have shown that geopolymers can form an impermeable layer under the action of fiber, which makes the geopolymer matrix structure denser [67]. The good adhesion between NCF and geopolymer matrix enables the composite material to prevent crack expansion, resist external environment erosion, and enhance its short- and long-term mechanical properties [68].

### 4.1. Influence of Different Kinds of Fibers on the Mechanical Properties of Geopolymers

Without considering the influence of environmental factors such as the wet–dry cycle on the composite materials, the mechanical properties of different NCFs are very different. Bast fiber has an obvious influence on the properties of geopolymers. Eyerusalem et al. [69] studied the fracture structure of hemp-fiber-reinforced geopolymer composites. There are fibers pulled out on the microscopic surface, which shows the toughening ability of the fibers. However, the compressive strength of the composite decreases continuously with the increase in fiber content, which indicates that the fiber distribution in the matrix is not uniform. Saez-Perez et al. [70] also conducted an experimental study on hemp fiber. They compared fresh hemp fiber with hemp fiber stored in a wet state for 6 months and found that wet preservation caused an increase in cellulose content and improved the mechanical properties of the geopolymer composite. Poletanovic et al. [71] found that hemp fiber had no significant effects on the density, water absorption, compressive strength, and bending strength of the composite material, but significantly increased the toughness and energy absorption capacity of the material. Trindade et al. [72] used sisal and jute single fibers as reinforcing materials and found that these fibers could change the brittleness of the matrix. Na et al. [73] used alkali-treated kenaf fiber as the geopolymer reinforcement, which improved the toughness of the composite by 473%. However, the study of Korniejenko et al. [74] also showed that the excessive addition of flax fiber would lead to the deterioration of the mechanical properties of composite materials. When flax fiber is added by 8%, the compressive strength of the material decreases by about 50%. The bending strength of pure geopolymer matrix is 3.45 MPa, and that of 8% flax-fiber-reinforced composite is 2.13 MPa, which further proves that the fiber content should not be too high.

Sisal fiber is essentially sisal leaf fiber. Ampol et al. [75] added sisal fiber and coconut fiber with different volume fractions into the geopolymer matrix, respectively, and compared the mechanical properties with glass-fiber-reinforced geopolymers. The results show that, compared with glass fiber, the bending strength of cellulose fiber is significantly improved, but the compressive strength of cellulose fiber has a tendency to decrease. Only appropriate fiber can improve the mechanical properties of geopolymers [76].

The most representative seed fiber is cotton fiber. Compared with pure geopolymers, the addition of cotton fiber gradually improves the fracture toughness of fiber-reinforced geopolymer composites. Tests have shown that even cotton fiber as a reinforcement can improve the bending properties of composite materials [77].

Fruit fiber refers to the fiber obtained from the fruit of the plant. It is mainly composed of cellulose and associated substances and cellular interstitium, such as coconut fiber. Zulfiat et al. [78] showed that the compressive strength and flexural strength of geopolymers reinforced with pineapple fiber with 0.5% mass fraction were higher than those of geopolymer reinforced with pineapple fiber with 0.5% mass fraction. However, excessive fiber will reduce the mechanical properties of composite materials. Kroehong et al. [79] showed that the increase in oil palm fiber content decreased the compressive strength of geopolymer, but improved the bending strength and toughness of the material, and changed the failure behavior of the composite. Mazen et al. [80] used loofah fibers as geopolymer reinforcement materials. Compared with pure geopolymers, the compressive strength of the composite was increased from 13 MPa to 31 MPa, and the bending strength was increased from 3.4 MPa to 14.2 MPa.

Wood fiber has also been used as a geopolymer reinforcement. Su et al. [81] compared lignin fiber, polypropylene fiber, and alkali-resistant glass-fiber-reinforced geopolymers. The formation and expansion of matrix cracks were improved by all fibers, and the strengthening effects of composites were polypropylene fiber, glass fiber, and lignin fiber, respectively. Gabriel et al. [82] increased the maximum added amount of wood fiber to 35 wt.%. When the fiber content is too large, the mechanical properties of the composite are inversely proportional to the fiber content.

The main purpose of adding fiber is not to improve the compressive properties of the matrix but to improve the bending strength of the composite and control the cracks of the matrix. Straw fiber includes rice, wheat, sorghum, bagasse, and other waste crop straw. Chen et al. [83] found that when the mass fraction of sweet sorghum fiber in geopolymers is less than 2.0%, the fiber content is negatively correlated with the compressive strength of the composite, while being positively correlated with the tensile strength and toughness of the composite.

### 4.2. Effect of Fiber Modification on Composite Materials

The toughness of composite materials is closely related to the fiber properties in the matrix. Compared with other conventional fibers, cellulose fibers degrade under natural conditions, which affects their properties. Modification treatment methods are generally used to improve the properties of cellulose fibers and increase the effective bonding with the contact surface of the geopolymer matrix [84,85].

Since the fibers are weakened by alkali erosion and mineralization of hydration products migrating to lumen and space, the bast fibers are usually modified by alkali treatment to improve the properties of composite materials [86]. Lazorenko et al. [87] treated flax tow with a 5% sodium hydroxide solution combined with ultrasound, which improved the properties of geopolymer composites.

Alkali treatment is also the main modification method for straw fiber [88]. Huang et al. [89] found that the interface bonding effect between alkali-treated straw fiber and geopolymer matrix was better than that of untreated fiber. Workiye et al. [90] treated corn stalk fiber with sodium hydroxide, indicating that the single fiber of corn stalk after alkali treatment improved the compressive strength of the geopolymer.

Ribeiro et al. [91] found that alkali treatment and water treatment can improve the toughness of bamboo fiber and bamboo strip-reinforced polymers, but have little difference in the bending strength of composite materials. After hot water washing, the specific strength of the pine particles geopolymer was increased by 27%, respectively [40]. The results showed that hot water could wash away the impurities of pine fiber and make the interfacial bond between ground fiber and ground polymer better. Sometimes acid treatment is also used. Roy et al. [92] soaked abaca fiber in an acidic solution of aluminum sulfate with a pH value of 6 and found that the tensile strength of acid-treated abaca fiber was higher than that of untreated and alkali-treated fibers.

The research shows that chemical treatment can improve the interface bond between geopolymer matrix and fiber, and the self-treatment behavior of the geopolymer can protect the fiber from the external environment. Self-treatment is the process by which fibers are directly added to the geopolymer slurry without any pretreatment process. Maichin et al. [93] studied the self-processing behavior of hemp fiber in geopolymer composites. The self-processing behavior of hemp fiber in geopolymers improves the final properties of the composite. Similarly, Maichin et al. [94] found that the fiber self-treatment process was affected by the alkalinity in the geopolymer environmental system. After the fiber is self-treated in the matrix, the fiber surface is modified to give the fiber stronger bonding properties.

## 5. Effect of Wet–Dry Cycles on Properties of Fiber-Reinforced Geopolymers

As structural materials of buildings, fiber-reinforced geopolymers are often exposed to varying wetting and drying environments during their service life cycle [58]. At present, the long-term properties of geopolymer materials are studied, including frost resistance, sulfate resistance, chloride resistance, heat resistance, etc. Among them, the influence of the wet–dry cycle on NCFRG composites is most closely related to engineering practice [95]. In the relevant research on geopolymers reinforced with NCFs under the condition of wet–dry cycling, the composition of the precursor, activator, and NCFs of composite materials is shown in Table 3. It is well known that the long-term properties of geopolymer composites are closely related to their compactness and crack resistance, and good toughness helps to improve the durability of geopolymers. Fiber can inhibit and stabilize the development of micro-cracks, which is an effective way to alleviate the deterioration of geopolymer composites.

### 5.1. Wet–Dry Cycle Test Method

In engineering practice, the method of the wet–dry cycle is usually adopted to accelerate the aging of building materials. Wet–dry cycle resistance is an important index of the long-term properties of composite materials. In the wet–dry cycle test, the size of the composite material specimen is generally carried out according to the relevant specifications of the test project. The pressure test generally adopts a cube specimen or prismatic specimen; The flexural test usually uses a thin plate or beam specimen, and the impermeability test uses a cylinder specimen. The curing conditions of NCFRG are generally standard curing conditions or room temperature conditions, and the curing time is generally 28 days. The wet–dry cycles vary slightly according to the test conditions and methods. The wet–dry cycles of some relevant studies were conducted by the relevant provisions of the code ASTMD599 [98,99]. Kamaruddin et al. [33] conducted wet–dry cycle tests after the curing period of 7, 28, and 90 days, respectively. They air-dried the samples for 24 h and then soaked them for another 24 h. A cycle lasts 48 h. The study used three different cycles for comparison, namely 1, 3, and 5 wet–dry cycles. Similar to most other studies, the duration of a wet–dry cycle ranges from 48 to 72 h [100,101]. The preparation of relevant specimens, curing conditions, and wet–dry cycles are shown in Table 4.

### 5.2. Effect of Wet–Dry Cycling on Geopolymer Matrix

Generally, the effects of various accelerated aging conditions on the matrix and fibers of composite materials can be studied [102,103]. The wet–dry cycle is one of the research methods to accelerate the aging of composite materials. Bui et al. [104] studied the deterioration mechanism and mechanical properties of mortar containing coconut fiber after repeated wetting and drying. The results show that although the compressive strength increases after the first cycle, the compressive strength and bending strength of the composite decrease significantly after the fifth cycle. Wei et al. [103] studied the influence of seven corrosion conditions on sisal-fiber-reinforced geopolymers, including dynamic wet–dry cycle conditions and static environments under different temperatures and humidity. It was found that the dynamic wet–drying cycle accelerated alkaline hydrolysis of amorphous components and cell wall mineralization.

#### 5.2.1. Influence of Wet–Dry Cycle on Compressive Strength of Composites

The number of wet–dry cycles is sensitive to the compressive strength of the material, and a small number of cycles will also have a greater impact on it. With the addition of fiber, the stress–strain curve of the composite changes from brittleness to toughness. Kamaruddin et al. [33] prepared fiber-reinforced geopolymers using soil, 60% fly ash as a precursor, lime and alkaline activator solution as an activator, and 1% coir fiber as reinforcement material. The influence of the number of wet–drying cycles during different curing periods on the compressive strength of the material was studied. It is found that the wet–dry cycle has a great influence on the compressive strength of lime samples and alkaline activator samples. After one wet–dry cycle and three wet–dry cycles, the compressive strength of lime samples and alkaline activator samples ranged from 1.41 to 1.88 MPa and from 2.64 to 8.29 MPa. In the five wet–dry cycles over 90 curing days, the compressive strength of lime and alkaline activator samples decreased from 1.62 to 1.25 MPa and from 6.06 to 5.89 MPa, respectively. The strength decline rates of lime and alkaline activator samples were 20.38% and 38.64%, respectively, after five wet–drying cycles over 90 days. Although the decline rate of the alkaline activator sample is higher, it is still better than that of the lime sample. This is since alkaline activators increase the intergranular interface binding of soil samples.

With the increase in the number of wet–dry cycles, the effect on the properties of composite materials becomes more and more obvious. Nkwaju et al. [95] quantified the resistance of bagasse-fiber-reinforced geopolymers to wet–dry cycle through experiments. It was found that the mass loss of the geopolymers increased with the increase in wet–drying cycles by observing the changes in sample strength and mass after different wet–drying cycles. After 5, 10, and 20 cycles, the mass loss of geopolymers ranges from 1.89 to 10.5%, 3.15 to 11.19%, and 3.07 to 13.01%, respectively. When the content of bagasse fiber is 1.5% and 3%, the resistance of the geopolymer to the wet–dry cycle is increased, and the mass loss of the composite is reduced. It can be seen that the 3% bagasse-fiber-reinforced geopolymer exhibits the lowest mass loss in all cycles. After 5, 10, and 20 cycles, the mass loss of the geopolymer reinforced with 3% bagasse fiber was 1.89%, 2.17%, and 3.07%, respectively. When the content of bagasse fiber was 1.5% and 3%, the resistance of the geopolymer to wet–dry cycle was enhanced, which was related to the effective bridging of geopolymer components at low dosages of bagasse fiber. On the other hand, compared with the pure bagasse fiber geopolymer, the mass loss is greater at 4.5, 6, and 7.5% of bagasse fiber content, as shown in Figure 4a. This is related to the increase in geopolymer pores. The increase in pores leads to more water entering, and the corresponding geopolymer is degraded, which is consistent with the results of Marvila et al. [105]. The effects of wet–dry cycling on the compressive strength of the geopolymer reinforced with bagasse fiber in different dosages are similar to the mass loss of the geopolymer, as shown in Figure 4b. The compressive strength loss of the geopolymer is positively correlated with the mass loss. It can be seen from the figure that 3% bagasse fiber content is the best content to improve the resistance of the geopolymer to the wet–dry cycle.

Nkwaju et al. [38] also studied the long-term properties of iron-rich red soil-base geopolymer composites made of bagasse fiber. The results show that the addition of fiber can improve the fracture property of geopolymers. When the fiber content increases from 0 to 7.5%, the 28-day compressive strength of the composite decreases from 50 MPa to 14 MPa. The ultimate compressive strength of geopolymer composites after 20 wet–dry cycles has no significant effect. With the increase in fiber content, although the compressive strength decreases, its ductility also increases. See Figure 5a,b. This is different from the failure mode results of cement-based materials after 20 wet–dry cycles of Puertas et al. [106]. The results show that cement-based materials have a much greater influence on the wet–dry cycle than geopolymers.

#### 5.2.2. Influence of Bending Strength

Fiber-reinforced geopolymer matrix composites are different from fiber-reinforced cement matrix composites. Under the condition of a wet–dry cycle, the chemical composition of fibers changes during the accelerated aging process, which improves the adhesion between fibers and the geopolymer matrix and enhances the adhesion between fibers and the geopolymer matrix [107].

Santos et al. [32] produced geopolymer composite materials using the sludge and sisal fiber generated from water treatment plants, and evaluated its bending strength through accelerated aging through 10 wet–dry cycles. In the bending test, the strength of 0 cycles of reference specimens reached 15 MPa, and the strength of 10 wet–dry cycles accelerated aging specimens was 11 MPa, showing good wet–dry cycle resistance. The fibers of the naturally aged 3-year composite samples showed little degradation.

Alomayri et al. [108] exposed the composites to 10, 20, and 30 wet–dry cycles to study the long-term properties of 10% cotton-fiber-reinforced geopolymers. The behavior of the composite in the inelastic stage, i.e., cracking mechanism, strength, and ductility, was tested by bending test. The results show that the slight degradation rate of the composites is as high as 30 cycles, and the bending strength is as high as 35 MPa. Wei et al. [42] found that the bending strength and toughness of fiber-reinforced cement mortar after 10 wetting and drying cycles decreased by 90% and 98%, respectively. After replacing 20% cement with natural mineral diatomite, the bending strength and toughness of the control group reached 5.3 times and 7.9 times, respectively, after 10 wetting and drying cycles. 

Huang et al. [96] studied the bending strength of fir-shaven fiber-reinforced geopolymers under different environments. The control specimens were those at the age of 28 days. Figure 6 shows the bending strength of indoor, outdoor, and bagged specimens at different test times and the change of the loss rate of the bending strength value with the test time. The bending strength of the composite material in indoor, outdoor, and bag environments decreases with time, but the amplitude is different, as shown in Figure 6a,b. On day 360, the bending strength loss rates of indoor, outdoor and bagged specimens were 21.1%, 47.1%, and 27.6%, respectively. It can be seen that in the outdoor environment, due to the wet–dry cycle of the external environment, as well as the common influence of temperature and sunlight, the matrix and fiber of the composite material are damaged, resulting in the decline in its bending strength.

The strength of the composite increases with the continuous polymerization reaction of the geopolymer with the aging time. Mazen et al. [79] found that the bending yield strength of loofah-fiber-reinforced geopolymers with 10% fiber content increased from 8.6 MPa to 9.8 MPa after 20 months of aging time under the condition of no obvious wet–drying cycle. During the 20-month aging period, the flexural strength and flexural modulus of the composite changed slightly, and the mechanical properties did not decrease significantly. 

Analysis shows that the addition of nanomaterials can improve the long-term properties of composites [109]. On the one hand, nanomaterials reduce the alkalinity of the system, thus reducing the degradation of fibers. At the same time, the nanomaterial accelerates the polymerization reaction, increases the amount and density of the polymer gel, and improves the adhesion between the fiber and the matrix. Assaedi et al. [110] studied the flexure strength of flax-fiber-reinforced geopolymers at 32 weeks of aging. Compared with 28-day flexural strength, the flexural strength of fiber-reinforced geopolymers without nanomaterials decreased by 22.4%, while the flexural strength of geopolymers containing nano-silica decreased by 10.3%. The results show that the bending strength of the nanocomposite decreases less. Ardanuy et al. [111] tested the mechanical properties of cellulose-nanofiber-reinforced mortar after five wet–dry cycles. The bending strength and modulus of cellulose-nanofiber-reinforced mortar are higher than those of sisal-fiber-reinforced mortar, but the fracture energy is lower than that of conventional sisal-fiber-reinforced mortar.

#### 5.2.3. Influence of Wet–Dry Cycle on Fiber

Keratinization mechanism of NCFs under wet–dry cycle

Cellulosic fiber keratosis refers to the irreversible removal of water from cell wall fibers through repeated wet–drying cycles [112]. The fibers are soaked in water and saturated and then dried. At this time, it was kept in the range of 60–80 °C [113,114]. The process of fiber drying is also the process of polysaccharide cellulose chain rearrangement. After rearrangement, the cellulose microfibril formed irreversible hydrogen bonds. When the fiber is wet–dry cycling again, the capillary cavity in the cell wall is closed due to the collapse of the fiber lumen, so that the fiber cannot return to the initial state of water absorption and expansion, so that the size of the fiber is stable [115]. It can be seen that the water absorption of NCFs after keratinization is reduced, thus weakening the degradation degree of cellulose fibers in the alkaline matrix environment. The dimensional stability improves the interface bonding between the fiber and the matrix, thus improving the performance of the composite [116,117,118].

Degradation behavior under wet–dry cycle conditions

In addition to the keratinization mechanism, cellulose fibers lose strength due to degradation when used to reinforce cement substrates exposed to environmental conditions of wet–dry cycles. Degradation of natural fibers in alkaline mineral environments exhausts the reinforcing effect of the fibers, resulting in reduced durability of the composite materials [119,120,121].

It is well known that the amorphous components of NCFs degrade in the alkaline environment of cement-based cementitious material substrates. Wei et al. [61] embedded sisal fiber into pure cement and metakaolin-modified cement matrix, and found that cement hydration was a key factor in fiber degradation behavior through a wet–dry cycle test. Metakaolin can reduce the alkalinity of pore solution and effectively alleviate the degradation of natural fibers. Ye et al. [122] also found that the geopolymer matrix and cellulose fiber had good bonding properties without significant degradation, while the alkaline degradation of hemicellulose reduced the degree of polymerization of geopolymers.

Yan et al. [123] verified that the performance of flax-fabric-reinforced composite material decreased most seriously in a 5% sodium hydroxide alkaline solution. After impregnating bamboo pulp cellulose fiber in cement concrete and geopolymers, Correia et al. [124] reduced the tensile strength of pulp by 70% and 34%, respectively, due to the degradation of hemicellulose and cellulose. Another similar study [92] also found that wood-fiber-reinforced geopolymers decreased their specific strength by 15% after repeated soaking and drying.

The high moisture absorption of natural fibers also creates favorable conditions for the biodegradation of polymer composites [125]. When more water is absorbed by the fiber, the bound water inside the fiber increases and the level of free water decreases. At this time, the leaching of water-soluble components in the fiber promotes the fiber to delaminate. Filho et al. [41] found that the bending behavior of sisal-fiber-reinforced cement-based composites changed significantly after 10 wet–dry cycles. The mechanical properties after 25 wet–dry cycles are the same as those of the unreinforced matrix. As for the geopolymer matrix, the fiber mineralization was not observed due to the low content of calcium hydroxide. Trindade et al. [40] after a 15 w/d wet–dry cycle of the jute-fabric-reinforced geopolymer, there was no significant change in the ultimate mechanical capacity of the composite material and no significant degradation of the fiber.

To analyze the degradation of fibers in the matrix, Mohr et al. [126,127] used 10% (MK-10) and 30% (MK-30) metakaolin instead of cement to conduct wet–dry cycle tests. Due to the presence of metakaolin, the initial first crack strength of kraft-pulp-fiber-reinforced mortar is improved. The MK-10 and MK-30 are 9% and 34% stronger than cement (PC), respectively. After 5 cycles, the first crack strength of PC increased slightly and then decreased with increasing wetting and drying cycles. Figure 7 shows the stress–strain curve of kraft pulp fiber. It can be seen that the tensile strength of the embedded fiber in MK30 after 5 wet–drying cycles is slightly higher than that of the fiber-fiber, while that after 30 wet–drying cycles is lower than that of the fiber-fiber, indicating the degradation of the kraft pulp fiber.

## 6. Influence of Wet–Dry Cycle on the Microstructure of Composites

The microstructure of the composite material can directly characterize its properties. Mohr et al. [128] studied the microstructure and chemical mechanism of pulp fiber-cement composite degradation during the wet–dry cycle through scanning electron microscopy and energy dispersion spectroscopy. A three-part progressive degradation mechanism of the current cast pulp fiber-cement composite was proposed. Cheng et al. [129] identified the pore structure characteristics of the samples under different corrosion environments through tomography and mercury injection pore method. It was found that the combination of sulfate and chloride salt slowed down the deterioration of mortar under a wet–dry cycle. The microstructure analysis showed that the sulfate ions in the chloride ion solution refined the pore structure, while the presence of sulfate ions and magnesium ions in the chloride ion environment increased the porosity and pore volume and accelerated the entry of corrosive ions. Kamaruddin et al. [33] showed through microscopic analysis that after 90 days of maintenance, the soil was dense and compacted, with fewer visible pores. The formed C-S-H and C-A-H gels help interlock between soil particles. Wei et al. [61] studied the alkali degradation process of natural fibers, including the hydrolysis of cellulose and hemicellulose, the stripping of cellulose microfibers, and the degradation of non-crystalline regions of the cellulose chain, in combination with the results of thermogravimetric analysis and microstructure.

Huang et al. [96] analyzed SEM images of the fracture slice morphology of control samples and 360-day indoor, outdoor, and bagged samples, as shown in Figure 8. On day 28, the fibers of the control sample were tightly bound to and embedded in the matrix, as shown in Figure 8a. Since the fibers in the control sample are only slightly attacked by the alkali at this time, the fibers are tightly bound to the matrix and therefore have a higher strength. Fibers were pulled out of the fractured surface of the laboratory sample that had been preserved for 360 days. The fiber surface was coated by the attachment, a part of the fiber was stripped and pulled out, and a part remained in the matrix, as shown in Figure 8b. The indoor samples at 360 days were mainly affected by the wet–dry cycle, and calcium hydroxide was formed due to hydration. Part of the hydrate migrated to the fiber cavity, wall, and void, resulting in fiber corrosion and mineralization, while the other part migrated to the sample surface with water, reducing the alkali erosion of the fiber. Therefore, although fiber strength is affected, the effect may be small. As shown in Figure 8b, no direct brittle fracture occurred in the fiber of the sample in the laboratory. Therefore, the fiber acts as a bridge and the specimen has a high strength. For the outdoor sample of 360 days, the fibers were not damaged by large external mechanical forces, as shown in Figure 8c. A small amount of matrix is attached to the surface of the fiber, one end of the fiber is embedded in the matrix, and there is a gap between the other end of the fiber and the matrix. Because the specimen is exposed to the most severe outdoor natural environment, the fiber corrosion is serious, the matrix is damaged most seriously, and the bending strength is reduced. It can also be seen from Figure 8c that the fiber is loosely bonded to the matrix, thus reducing the contact surface between the fiber and the matrix, reducing the stress transfer efficiency, and aggravating the reduction in the bending strength of the specimen. Therefore, the bending strength of outdoor specimens is the lowest. The bagged specimens are less affected by the wet–dry cycle; thus, the alkalinity remains unchanged. Because the fiber is embedded, mineral ion Ca^2+^ is more likely to deposit on the fiber surface, which easily leads to fiber mineralization, resulting in ductile fibers becoming brittle. As shown in Figure 8d, the fibers on the surface of the bag sample were broken, and the residual fibers were still tightly embedded in the matrix, indicating that in this case, the fiber strength decline led to the decline in the bending strength of the sample, making the strength of the bag sample lower than that of the indoor sample. The above analysis results show that the outdoor samples are seriously affected by the climate, resulting in poor fiber–matrix adhesion and significant strength loss. The bagged sample was seriously corroded by alkali and the strength loss was great.

Correia et al. [124] used X-ray photoelectron spectroscopy and Fourier transform infrared spectroscopy to analyze the changes in the composition and chemical structure of pulp fibers soaked in different pulps. The results showed that the immersion of pulp and nanocellulose in geopolymers and cement slurry changed the chemical surface of these fibers. The lignin, hemicellulose, and cellulose containing a certain degradation were removed. After soaking in cement and geopolymers, the tensile strength of the pulp sheets decreased by 70% and 34%, respectively. Analysis of pulp immersed in different mucilages also showed a change in the chemical structure of the fibers when exposed to an alkaline environment.

The fiber length and its distribution in the matrix also affect the properties of the composite. Zulfiat et al. [78] tested the addition of pineapple fiber geopolymers with lengths of 10 mm, 20 mm and 30 mm. It is found that the compressive strength and bending strength of the composite with fiber content of 0.50% and fiber length of 30 mm are 41.468 MPa and 9.209 MPa, respectively. The fiber length with compressive and flexural strength values of 30 mm is higher than that of 10 mm and 20 mm. In theory, long fibers can transfer stress and load from the stress point to another fiber. The fiber length of 30 mm is evenly distributed, which makes the binding effect between the pineapple-fiber-reinforced fiber and the matrix good.

## 7. Future Directions and Limitations

Under the condition of wet–dry cycles, the influence of different types of NCFs is very different. This can be confirmed by the hygroscopicity of the fibers. At 97% relative humidity, the moisture absorption rate of water hyacinth fiber is almost 8 times higher than that at 75% relative humidity. Under the same conditions, the moisture absorption rate of sisal, reed, and rose leaf fibers only increased by 4% [65]. Another study found that coir fiber has a maximum moisture absorption rate of 180%. Under the same conditions, the maximum moisture absorption rate of jute fiber is only 40% [60]. The same fiber will also have different moisture absorption rates due to differences in the external environment. The moisture absorption rate of sisal fiber soaked in water is 5% lower than that of sisal fiber soaked in lime water [61].

According to the above analysis, the high hygroscopicity of NCFs limits the resistance of composites to wet–dry cycles. In order to reduce the influence of wet–dry cycles on the properties of composites, further optimization measures should be taken in future engineering practice.

NCFs with a low moisture-absorption rate, such as sisal fiber, jute fiber, etc., should be used.

Alkali pretreatments should be used to improve the resistance of NCFs to wet–dry cycles.

Nanomaterials are added to composites. The nanomaterials in the matrix can reduce the alkalinity and slow down the degradation degree of the NCFs. Nanomaterials can also accelerate the polymerization reaction, increase the amount and density of the polymer gel, and improve the adhesion between the NCFs and the matrix.

The keratinization mechanism of NCFs can also be exploited. The water absorption of NCFs after keratination is reduced, thus reducing the degradation of NCFs in the alkaline matrix environment.

## 8. Conclusions

Although there are a lot of studies on NCFRG composites, there are few studies on the durability of cellulose-fiber-reinforced geopolymer composites which are important for reliability under wet–dry cycle conditions. In particular, research on the aging of composite materials with engineering practical significance based on the natural environment is less. In this paper, the wet–dry cycle resistance of NCFRG is summarized and analyzed, which lays a foundation for its engineering application. The main conclusions and prospects are as follows:

The alkaline degradation of cellulose fibers in the geopolymer matrix has adverse effects on the mechanical properties of composites, which can be mitigated by chemical modification and self-modification.

The effects of the wet–dry cycle on composite materials include mechanical properties and long-term properties, which have a greater impact on long-term properties. It can improve the compactness of the matrix and improve the long-term performance of the composite under the condition of wet–dry cycle resistance.

The influence of the wet–dry cycle on composite materials includes the influence on matrix and the influence on fiber, and the influence on fiber is greater. The effects on fiber include fiber keratosis and fiber alkaline degradation.

The appropriate amount of cellulose fiber has a favorable effect on the mechanical properties, toughness, and cracking resistance of the geopolymer, and too much will have a reverse effect. The results showed that 3% bagasse fiber content was the best content to improve the resistance of geopolymers to the wet–dry cycle.

For NCFRG composites, the influence of wet–dry cycling on fibers is highlighted by the change in fiber microstructure, which is reflected in fiber degradation and mineralization.

## Figures and Tables

**Figure 1 molecules-28-07189-f001:**
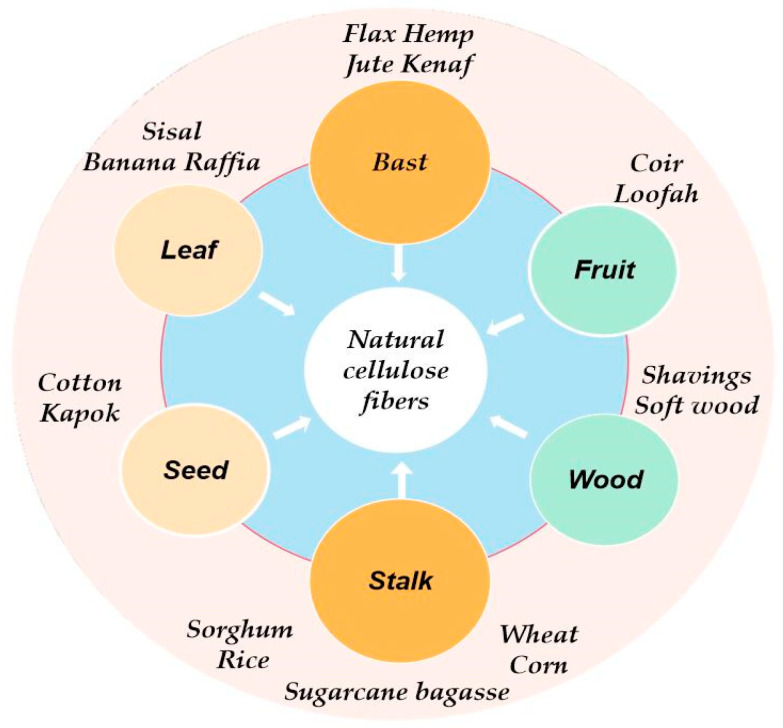
Classification of NCFs.

**Figure 2 molecules-28-07189-f002:**
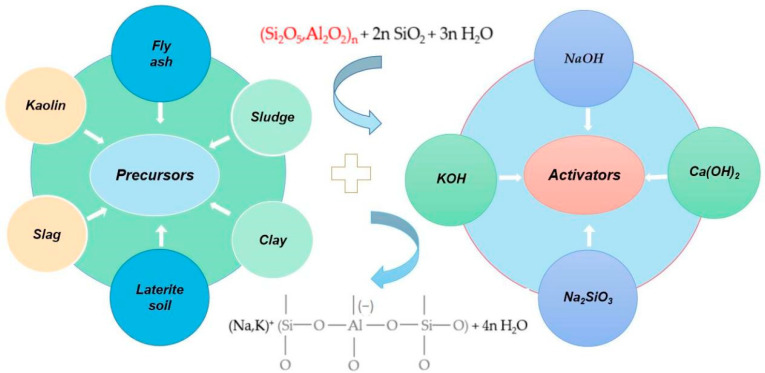
The Schematic diagram of the main components of different geopolymers.

**Figure 3 molecules-28-07189-f003:**
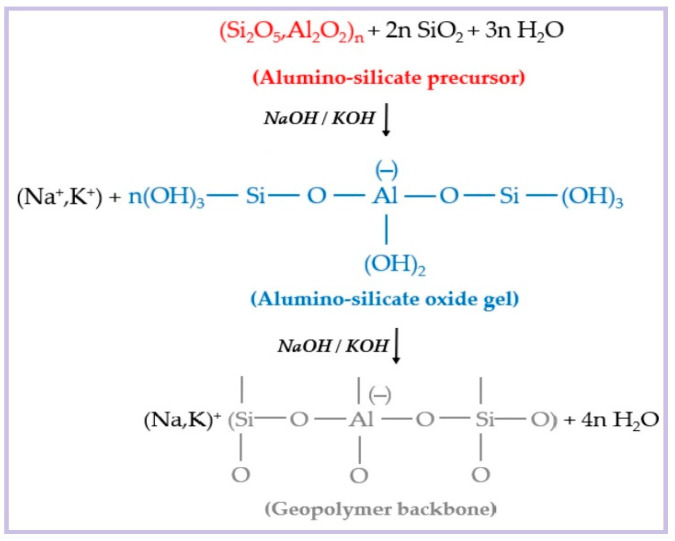
The polymerization process of geopolymers.

**Figure 4 molecules-28-07189-f004:**
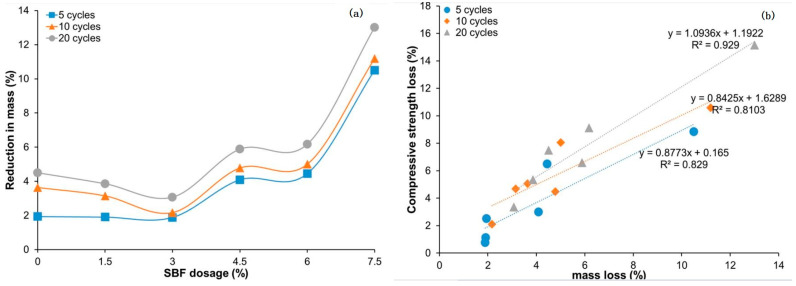
Effect of different dosages of bagasse fiber on properties of geopolymers in the wet–dry cycle. (**a**) Mass loss; (**b**) relationship between mass loss and compressive strength loss [95].

**Figure 5 molecules-28-07189-f005:**
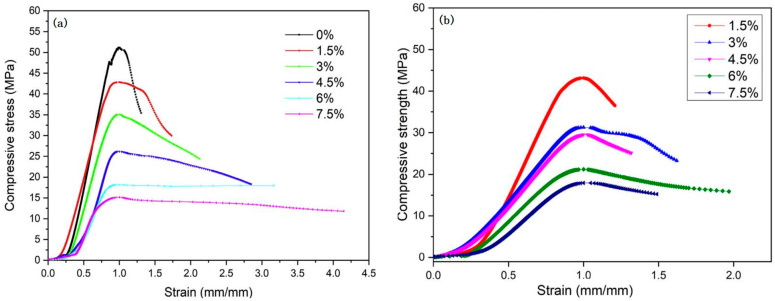
Geopolymers with different fiber content after 20 wet–dry cycles. (**a**) Ultimate compressive strength; (**b**) deformation capacity [38].

**Figure 6 molecules-28-07189-f006:**
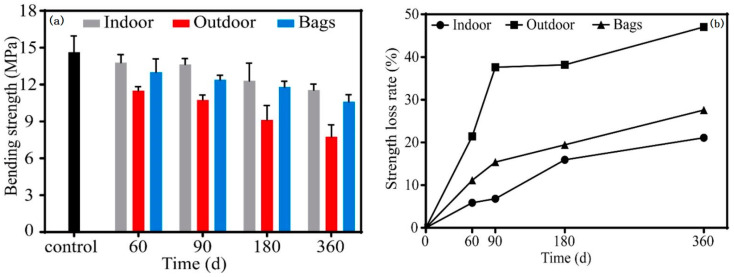
Flexural properties of indoor, outdoor, and bagged specimens at different test times. (**a**) Flexural strength; (**b**) loss rate of bending strength value [96].

**Figure 7 molecules-28-07189-f007:**
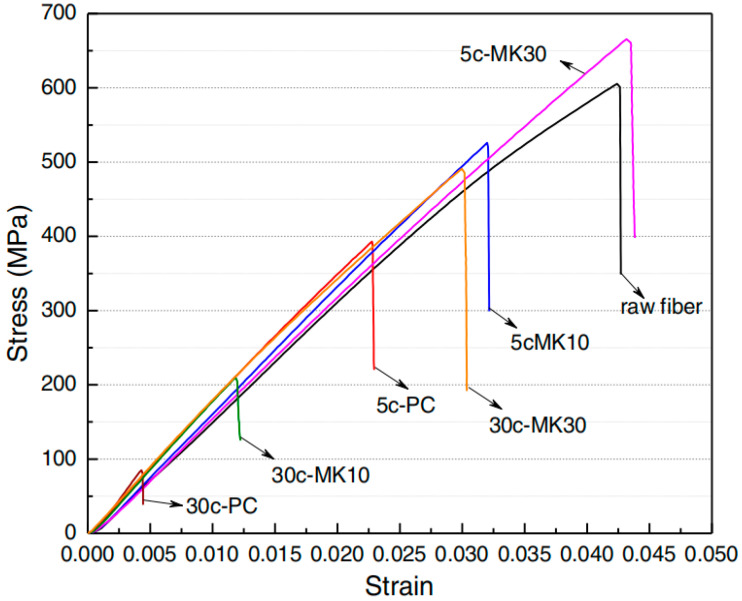
Stress–strain curve of tensile test on sisal fibers [126].

**Figure 8 molecules-28-07189-f008:**
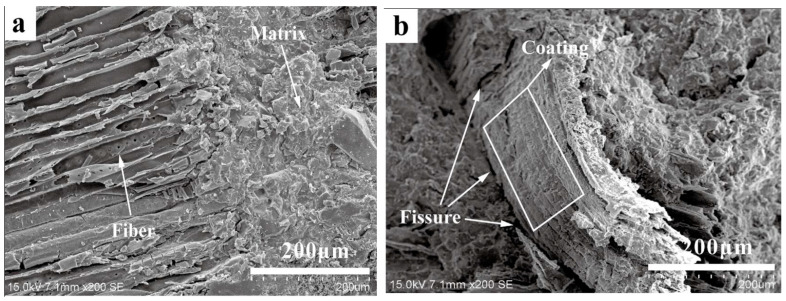
SEM images of composite surface. (**a**) 28-day control sample; (**b**) 360-day laboratory samples; (**c**) 360-day outdoor sample; (**d**) 360-day bag sample [96].

**Table 1 molecules-28-07189-t001:** Mechanical properties of some NCFs.

Fiber	Density /(g·cm^−3^)	Tensile Strength/MPa	Modulus /GPa	Elongation /%	Ref.
Flax	1.40–1.50	2200	27.6–80.0	3.7	[27,28]
Hemp	1.48	550–900	70.0	2.0–4.0	[29]
kenaf	1.45	930	53.0	1.6	[30]
Sisal	0.90	577	19.0	3.0–7.0	[31,32]
Banana	1.35	600	17.9	3.4	[31]
Coir	1.20	60–130	4.0–6.0	30.0	[31,33]
Cotton	1.60	400	4.8	7.0–8.0	[34,35]
Bagass	1.30	222–290	17.0–27.0	1.1	[36]

**Table 2 molecules-28-07189-t002:** The chemical composition of some NCFs.

Fiber	Cellulose/%	Hemicellulose/%	Lignin/%	Others/%	Ref.
Sisal	54.0–66.0	12.0–17.0	7.0–14.0	2.0–8.0	[40]
Jute	72.0	12.8	8.1	7.1	[41]
Coir	9.5–71.5	17.0–17.8	4.4–4.7	4.0–4.5	[42]

**Table 3 molecules-28-07189-t003:** Composition of NCFRGs under wet–dry cycle conditions.

Precursor	Activator	Fiber	Fiber Content/%	Ref.
Sludge and Portland cement	Potassium hydroxide solution	Sisal	2	[32]
Soil, hydrated lime, and fly ash	Potassium hydroxide solution	Coir	1.0	[33]
Iron-rich laterite	Sodium hydroxide and Sodium silicate solution	Sugarcane bagasse	1.5, 3.0, 4.5, 6.0 and 7.5 *	[38]
Fly ash	Sodium hydroxide and Sodium silicate solution	Wood particle	20 *	[92]
Metakaolin and cement	Polycarboxylate and Superplasticizer	Sisal	2	[58]
Metakaolin, silica fume, and blast furnace slag	Sodium hydroxide and Sodium silicate solution	Jute	10	[40]
Metakaolin and cement	Superplasticizer	Sisal	6	[41]
Diatomaceous earth and Portland cement	Limestone	Sisal	2	[42]
Laterite soil	Sodium hydroxide and Sodium silicate solution	Sugarcane bagasse	1.5, 3.0, 4.5, 6.0 and 7.5 *	[95]
Slag	Sodium hydroxide and Sodium silicate solution	Shaving	13 *	[96]
Silt, clay, and fly ash	Sodium hydroxide and Sodium silicate solution	Wastepaper	10, 20 and 30	[97]

The * indicates the mass content, the other is the volume content.

**Table 4 molecules-28-07189-t004:** NCFRG curing and wet–dry cycle conditions [32,33,38,41,42,61,92,95,96,97].

Author	Specimen Shape	Specimen Size/mm	Curing Condition	Curing Time/d	Cyclic Mode	Duration of A Cycle/h	Cycle Number
Kamaruddin et al. [33]	Cylinder	Diameter 50, Height 100	Standard curing chamber	7, 28 and 90	Air dry for 24 h, then soak for 24 h	48	1, 3 and 5
Nkwaju et al. [38]	Cube; Cylinder:	50 × 50 × 50; Diameter 100, Height 50	Room condition	28	Soak in water for 24 h, then dry in the open air for 24 h	48	5, 10 and 20
Batista dos Santos et al. [32]	Plate	195 × 50 × 6	27 °C and 80% RH	180	Soak in water at 22 ± 2 °C for 1 d, dry in an oven at 40 ± 2 °C for 2 d	72	10
Huang et al. [96]	Plate	220 × 50 × 10	Constant temperature drying oven, 40 °C, 28 d	28, 60, 90, 180 and 360	Indoor, outdoor, and bagged (indoor 11–32 °C)	-	Random
Filho et al. [41]	Plate	400 × 100 × 12	Curing chamber with 100% RH and 23 ± 1 °C	28 to 5 years	Soak in water 1 d, 2 d in a forced air chamber	72	5, 10, 15, 20 and 25
Wei et al. [42]	Plate	200 × 50 × 12	Lime water at 23.2 °C	28	Soak in a stainless steel container for 25 h and dry in a ventilated oven at 70 °C for 35 h	60	5, 10 and 20
Nkwaju et al. [95]	Prism; cylinder	40 × 40 × 160; Diameter 50, Height 150	20 ± 2 °C, store in plastic bag	28	Dry at 60 °C for 24 h, then soak in water for 24 h	48	5, 10 and 20
Wei et al. [61]	Beam; Cube	200 × 50 × 12; 50 × 50 × 50	Soaked in saturated water at 23 ± 2 °C	28	Soak in 70 °C water for 15 h and dry in 70 °C oven for 25 h.	42	5, 15 and 30
Trindade et al. [40]	Plate	450 × 60 × 12, and 270 × 60 × 12	Room temperature (25 ± 2 °C)	7	24 h of wetting and 48 h of drying	72	15
VU et al. [97]	Cylinder	Diameter 50, Height 100	Wrapped in vinyl sheet, 20 ± 3 °C	28 and 42	2 d in 40 °C electric furnace, 1 d in 20 °C water	72	10
Asante et al. [43]	Prism	25 × 30 × 50	Stored at 20 °C, 65% RH	7	Soak in water at 20 ± 5 °C for 170 min, in a ventilated oven at 70 ± 5 °C for 170 min	6	200

RH—relative humidity; h—hour(s); d—day(s).

## Data Availability

Not applicable.

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
