# Peer review of "Effect of Wet–Dry Cycling on Properties of Natural-Cellulose-Fiber-Reinforced Geopolymers: A Short Review"

_molecules, 2023, doi:10.3390/molecules28207189_

Round 1
Reviewer 1 Report
Chun Lv with collegueas have writed a review "Effect of wet-dry cycling on properties of natural cellulose fiber 2 reinforced geopolymer" compiling the available information on natural cellulose fiber-reinforced geopolymers and their indurance during wet-dry cycling. However, there are inconsistencies in writing and formating that need to be addressed as well as holes in the review concept.
-) Key words should be reevaluated (the ones mentioned in the title should not be in keywords as well).
-) Page 2, row 50: NCFs should be decifered: each new section demands decifering abreviations - natural celullose fibers (NCFs) are deciferend in abstract but not in introduction (samme for NCFRG in row 58)
-) Abstract and introduction last 2 centences are the same, this distracts the reader - one section ending is suggested for rewriting.
-) Table 1. Ref. should be formated same as in text [27,28] etc. (same for Table 2.)
-) ROw 104-107 NCFs are compared to other fibers. Higher elongation at break in comparison to what? Atleast one example should be given.
-) "By improving the interface 151 bonding state between the geopolymer matrix and NCF, and optimizing the bonding 152 characteristics of the interface layer between the geopolymer matrix and NCF, the com-153 posite material can achieve the best bonding performance [49,50]." This centence in rows 151-154 is to long and hence confuzes the reader. Centence should be optimezed to something in the direction as given example: "By improving the interface bonding state and optimizing the bonding characteristics between the geopolymer matrix and the NCF, composite material can achieve improved bonding performance [49,50]." ("best bonding performance" very strong statement - what is best?)
-) Table 3. spacings should be checked: example "2 ℃" and "2℃", same as "200×50×12" and "40x40x160" Speciment size column ("x") variations.
-) Spacing should be checked through out the whole text the are inconsistencies for example: "15 MPa" and "35MPa".
-) Figure 3. - in the figure there are no (a) and (b) atributed to the images (same for Figure 4 and 5)
-) "60-80 C" page 13 row 472
-) When reinforced composites are considered renforcement dimensions and size distribution is of extreem importance, however, authors have not discussed this aspect at all.
-) Different sources of natural cellulose fibers and their different properties, performance are discussed, however, chemical composition of those fibers should at least be mentioned in a table, mainly due to fact that it directly affects the interface adhesion with the chousen matrix.
Some centences are to long and complicated for easy understanding while reading.
Author Response
-) Key words should be reevaluated (the ones mentioned in the title should not be in keywords as well).
Thank you very much for reviewing our manuscript in your busy schedule. Based on your valuable suggestions, we have carefully revised the key words. (The revised page 1, lines 28-29).
-) Page 2, row 50: NCFs should be decifered: each new section demands decifering abreviations - natural celullose fibers (NCFs) are deciferend in abstract but not in introduction (samme for NCFRG in row 58).
Thank you very much for your review. According to your suggestion, We have deleted the sentence. (The revised page 2, lines 51-58).
-) Abstract and introduction last 2 centences are the same, this distracts the reader - one section ending is suggested for rewriting.
Thank you very much for your review. According to your suggestion, we have carefully revised our manuscript. (The revised page 1, lines 26-27).
-) Table 1. Ref. should be formated same as in text [27,28] etc. (same for Table 2.).
Thank you very much for your review. According to your suggestion, we have carefully revised our manuscript. (The revised page 3, line 106; The revised page 8, line 343).
-) ROw 104-107 NCFs are compared to other fibers. Higher elongation at break in comparison to what? Atleast one example should be given.
Thank you very much for your review. According to your suggestion, we have carefully revised our manuscript. (The revised page 3, lines 108-110)
-) "By improving the interface bonding state between the geopolymer matrix and NCF, and optimizing the bonding characteristics of the interface layer between the geopolymer matrix and NCF, the composite material can achieve the best bonding performance [49,50]." This centence in rows 151-154 is to long and hence confuzes the reader. Centence should be optimezed to something in the direction as given example: "By improving the interface bonding state and optimizing the bonding characteristics between the geopolymer matrix and the NCF, composite material can achieve improved bonding performance [49,50]." ("best bonding performance" very strong statement - what is best?).
Thank you very much for your review. According to your suggestion, We have deleted and corrected this sentences (The revised page 5, lines 165-168).
-) Table 3. spacings should be checked: example "2 ℃" and "2℃", same as "200×50×12" and "40x40x160" Speciment size column ("x") variations.
Thank you very much for your review. According to your suggestion, we have carefully revised the content. (The revised page 9, line 363).
-) Spacing should be checked through out the whole text the are inconsistencies for example: "15 MPa" and "35MPa".
Thank you very much for your review. According to your suggestion, we have carefully revised the content. (The revised page 12, line 449; The revised page 13, line 472;).
-) Figure 3. - in the figure there are no (a) and (b) atributed to the images (same for Figure 4 and 5).
Thank you very much for your review. According to your suggestion, we have carefully revised the content. (The revised page 11, lines 419-420; The revised page 12, lines 431-432; The revised page 13, lines 467-468).
-) "60-80 C" page 13 row 472.
Thank you very much for your review. According to your suggestion, we have carefully revised the content. (The revised page 13, line 494).
-) When reinforced composites are considered renforcement dimensions and size distribution is of extreem importance, however, authors have not discussed this aspect at all.
Thank you very much for your review. According to your suggestion, we have carefully revised the content. (The revised page 17, lines 615-623).
-) Different sources of natural cellulose fibers and their different properties, performance are discussed, however, chemical composition of those fibers should at least be mentioned in a table, mainly due to fact that it directly affects the interface adhesion with the chousen matrix.
Thank you very much for your review. According to your suggestion, we have carefully revised the content. (The revised pages 3-4, lines 120-124).
In addition, we have also revised other parts of the article according to your review suggestions (highlighted parts in the manuscript).
According to your suggestion, we have carefully and comprehensively revised the manuscript.
Finally, thank you again for your wonderful review of our article in your busy schedule.
Reviewer 2 Report
The paper "Effect of wet-dry cycling on properties of natural cellulose fiber reinforced geopolymer" was brilliantly presented. The title is clear. The abstract provided all the necessary information.
Refferences follow research. The methods are clearly presented. This work shows the innovativeness of geopolymer materials. A proposal to continue the research with changes in molarity. Kudos to the research and the research team.
Author Response
Thank you very much for your review. According to your suggestion, we have carefully revised the content.
In addition, we have also revised other parts of the article according to your review suggestions (highlighted parts in the manuscript).
Finally, thank you again for your wonderful review of our article in your busy schedule.
Reviewer 3 Report
The manuscript entitled "Effect of wet-dry cycling on properties of natural cellulose fiber-reinforced geopolymer" presents a brief overview of the literature regarding the use of natural fibers in geopolymer composition. However, the paper has a few issues that must be addressed. The paper needs minor revisions before it is processed further. Some comments follow:
The title doesn’t reflect the content of the paper. Also, please give the terms a short review.
Abstract. Please highlight the novelty of the study. The abstract is written qualitatively. The influence of the studied parameters should be presented in key sentences that present their negative or positive influence or their oprimal value. This section must be suitable for separate or independent publication. Currently, the formulations "shortly analyzed”, "summarized”, "briefly described”, "analyzed," and "promised" look redundant.
The introduction section should be improved. Please introduce correlating citations to assure a clear correlation between the cited study and the presented information. For example, in lines [8–10], only two citations were introduced for a multitude of parameters and conditions previously presented in the literature. What about mine tailings?
Figure 2 does not present the reaction mechanism of geopolymers; that is more likely a schematic representation of their ontaining. The reaction mechanism should include and describe all geopolymerization stages. Currently, the figure is more artistic than scientifically presented.
Please introduce some tables and figures that can summarize the content of sections 2-4. Currently, the information is hard to follow.
Also, as the authors stated and summarized in Table 1, there are multiple types of natural fibers. Therefore, the authors present only the effect on the properties of the geopolymers of some of the fibers (in sections 3-6). The authors should reorganize the paper, considering the effect of each type of fiber, and then compare and make relevant affirmations regarding which type of fiber is strongly affected by these cycles and which properties are affected and which aren't.
Future directions and limitations: Please provide some future directions and limitations of the study. This section is very important because the paper is a review type of study.
Author Response
- The title doesn’t reflect the content of the paper. Also, please give the terms a short review.
Thank you very much for reviewing our manuscript in your busy schedule. Based on your valuable suggestions, we have carefully revised the title. (The revised page 1, lines 1-2).
- Abstract. Please highlight the novelty of the study. The abstract is written qualitatively. The influence of the studied parameters should be presented in key sentences that present their negative or positive influence or their oprimal value. This section must be suitable for separate or independent publication. Currently, the formulations "shortly analyzed”, "summarized”, "briefly described”, "analyzed," and "promised" look redundant.
Thank you very much for reviewing our manuscript in your busy schedule. Based on your valuable suggestions, we have carefully revised our manuscript.
( the revised page 1, lines 19-27)
- The introduction section should be improved. Please introduce correlating citations to assure a clear correlation between the cited study and the presented information. For example, in lines [8–10], only two citations were introduced for a multitude of parameters and conditions previously presented in the literature. What about mine tailings?
Thank you very much for reviewing our manuscript in your busy schedule. Based on your valuable suggestions, we have carefully revised our manuscript. (The revised page 2, lines 47-49; The revised page 19, lines 704-705).
- Figure 2 does not present the reaction mechanism of geopolymers; that is more likely a schematic representation of their ontaining. The reaction mechanism should include and describe all geopolymerization stages. Currently, the figure is more artistic than scientifically presented.
Thank you very much for reviewing our manuscript in your busy schedule. Based on your valuable suggestions, we have carefully revised our manuscript. (The revised page 4, lines 139-140).
- Please introduce some tables and figures that can summarize the content of sections 2-4. Currently, the information is hard to follow.
Thank you very much for reviewing our manuscript in your busy schedule. Based on your valuable suggestions, we have carefully revised our manuscript. (The revised page 4, line 124;The revised page 4, lines 142-145; The revised page 5, lines 159-160).
- Also, as the authors stated and summarized in Table 1, there are multiple types of natural fibers. Therefore, the authors present only the effect on the properties of the geopolymers of some of the fibers (in sections 3-6). The authors should reorganize the paper, considering the effect of each type of fiber, and then compare and make relevant affirmations regarding which type of fiber is strongly affected by these cycles and which properties are affected and which aren't.
Thank you very much for reviewing our manuscript in your busy schedule. Based on your valuable suggestions, we have carefully revised our manuscript. (The revised page 17, lines 625-634).
- Future directions and limitations: Please provide some future directions and limitations of the study. This section is very important because the paper is a review type of study.
Thank you very much for reviewing our manuscript in your busy schedule. Based on your valuable suggestions, we have carefully revised our manuscript. (The revised page 17, lines 635-648).
In addition, we have also revised other parts of the manuscript according to your review suggestions (highlighted parts in the manuscript).
According to your suggestion, we have carefully and comprehensively revised the manuscript.
Finally, thank you again for your wonderful review of our article in your busy schedule.
Round 2
Reviewer 1 Report
The Authors accepted quite all the suggestions, however, some few minor editions are still required:
-) English language edditing necessary: Page 1, row 18 “In this paper,” written twice
-) Key words still contain the same words as the Title “wet-dry cycling, geopolymer”, in the Title there is “fiber reinforced” in key words “fiber-reinforced” – no consistency.
After these things will be addressed by the authors I beliewe this manuscript can be accepeted for publishing.
Minor english language check needed.
Author Response
-) English language edditing necessary: Page 1, row 18 “In this paper,” written twice.
Thank you very much for your review. According to your suggestion, we have carefully revised the content. (The revised page 1, line 19).
-) Key words still contain the same words as the Title “wet-dry cycling, geopolymer”, in the Title there is “fiber reinforced” in key words “fiber-reinforced” – no consistency.
Thank you very much for your review. According to your suggestion, we have carefully revised the content. (The revised page 1, line 28).
Reviewer 3 Report
Dear Authors,
You have done a great job revising the paper. I don't have any additional recommendations that can improve your paper.
Best regards,
Author Response
Thank you for your wonderful review of our article in your busy schedule.